# Baseline Optical Coherence Tomography Parameters That May Influence 6 Months Treatment Outcome of Polypoidal Choroidal Vasculopathy Eyes with Combination Therapy: A Short-Term Pilot Study

**DOI:** 10.3390/ijerph18105378

**Published:** 2021-05-18

**Authors:** Rituparna Ghoshal, Sharanjeet Sharanjeet-Kaur, Norliza Mohamad Fadzil, Somnath Ghosh, NorFariza Ngah, Roslin Azni Abd Aziz

**Affiliations:** 1Optometry & Vision Science Program, Faculty of Health Sciences, Universiti Kebangsaan Malaysia, Jalan Raja Muda Abdul Aziz, Kuala Lumpur 50300, Malaysia; rituparna4ab@yahoo.co.in (R.G.); norlizafadzil@ukm.edu.my (N.M.F.); 2Department of Optometry, C T University, Ferozepur Road, Sidhwan Khurd 142024, Ludhiana, Punjab, India; 3Department of Allied Health Sciences, Brainware University, Barasat, Kalkata 700125, West Bengal, India; somnath4ab@yahoo.co.in; 4Department of Ophthalmology, Hospital Shah Alam, Persiaran Kayangan, Seksyen 7, Shah Alam 40000, Malaysia; drfarizangah@gmail.com (N.F.N.); roslinazni@gmail.com (R.A.A.A.)

**Keywords:** combination therapy, polypoidal choroidal vasculopathy, optical coherence tomography

## Abstract

Although optical coherence tomography (OCT) parameters have assisted in the diagnosis of polypoidal choroidal vasculopathy (PCV), its potential to evaluate treatment outcomes has not been established. The purpose of this pilot study was to evaluate baseline OCT parameters that may influence treatment outcome in PCV eyes with combination therapy. In this single-centered, prospective study, patients were recruited with at least one treatment-naïve PCV eye and treated with combination therapy of intravitreal anti-vascular endothelial growth factor and photodynamic therapy. Best-corrected distance and near visual acuity (DVA and NVA), and contrast sensitivity (CS) were recorded at baseline and six months after treatment. OCT parameters were determined. Twenty-six eyes of 26 patients aged between 51 to 83 years were evaluated. In eyes that had disrupted external limiting membrane (ELM), photoreceptors inner and outer segment (IS-OS) junction at 1000 micron of fovea at baseline showed low mean visual functions after 6 months of treatment. Eyes with foveal sub-retinal fluid (SRF) and polyp at central 1000 micron of fovea at baseline showed significantly worse DVA and CS after six months. Thus, the presence of foveal SRF, foveal polyp, disrupted ELM, and IS-OS junction at baseline significantly influenced the six months’ visual outcome in PCV eyes treated with combination therapy.

## 1. Introduction

Ophthalmic imaging has been revolutionized by the introduction of optical coherence tomography (OCT). It uses an interferometric imaging technique that produces cross-sectional images by mapping reflections of low-coherence laser light from tissue based on its depth. From the time of its inception, OCT has been efficiently attempting to provide minute clinical measurements of retinal layers in an automated fashion. With OCT, imaging of a multilayer retina represents the depth of each layer with the amplitude of spectrum modulation that is proportional to the specific reflectivity of the layer. Thus, OCT plays a major role in the diagnosis and management of a broad range of retinal diseases enabling a detailed evaluation of conditions including age-related macular degeneration, central serous retinopathy, retinal vein occlusion, diabetic retinopathy, and inherited retinal diseases [1]. With time, newer approaches towards identifying different OCT-derived parameters have been tested for retinal pathologies. Eventually, several OCT-based qualitative and quantitative parameters have been adopted in the diagnosis, treatment, and monitoring of different retinal diseases including different subtypes of neovascular age-related macular degeneration (AMD) [2,3,4].

Polypoidal choroidal vasculopathy (PCV) is a commonly seen subtype of neovascular age-related macular degeneration (n-AMD) in the Asian population including the Malaysian population [5,6]. It is characterized by multiple pigment epithelium detachments, polypoidal lesions, and branched vascular network [3]. Intravitreal anti-vascular endothelial growth factor (anti-VEGF) and photodynamic therapy (PDT) are the mainstays of treatment on PCV eyes. Visual functions, retinal morphology, and quality of life significantly improve with treatment in PCV eyes [7,8]. While previous research has enriched our knowledge on different OCT parameters that can guide clinicians in managing typical n-AMD, there is limited literature in this regard in eyes with PCV. Research data on the association between post-treatment visual functions and baseline OCT parameters in PCV eyes are limited. The majority of the related research derived indocyanine green angiography (ICGA)-based predictive factors like size, location, the texture of the polyp, and the greatest linear dimension (GLD) [9,10]. Considering the limited availability of ICGA in ophthalmic set-ups, OCT-based diagnosis criteria of PCV have already been recognized [11]. However, a detailed evaluation of OCT-based parameters that have the potential to influence the treatment outcome of PCV eyes has yet not been clearly established. Thereby, the purpose of the present study was to evaluate baseline OCT parameters that may influence treatment outcome in PCV eyes with combination therapy. This was achieved by determining the association between the six months after-treatment visual functions and baseline qualitative and quantitative OCT parameters.

## 2. Materials and Methods

### 2.1. Study Design and Treatment 

A longitudinal prospective pilot study was conducted in the Ophthalmology Clinic of a Public Hospital of Malaysia. A universal sampling method was used for the study. Patients with PCV who were advised to undergo combination therapy with intravitreal ranibizumab (anti-VEGF) and PDT were included in this study. PCV patients undergoing other treatments for any retinal pathology other than PCV and any history of previous treatment (anti-VEGF injection, PDT, or laser) for PCV were excluded. Similarly, patients with a larger lesion (lesion size > 5400 μm with ICGA), multiple polyps, foveal fibrosis, and thinned fovea (central retinal thickness < 200 μm with OCT) wherein PDT is not advisable, were excluded [12]. Furthermore, eyes with significant media opacity were also excluded. All patients underwent comprehensive eye examination and ophthalmic imaging including FFA, OCT, and ICGA. Senior retinal consultants (N.F.N, R.A.A.A.) evaluated all the subjects in detail before recruitment. The diagnosis of PCV has been described in our earlier paper [3]. In this pilot study, all patients with PCV who were seen in the ophthalmology department of this public hospital from December 2016 to March 2017 and who met the inclusion and exclusion criteria of the study were recruited. All patients included in this study received the first dose of intravitreal anti-VEGF (0.5 mg ranibizumab) (Lucentis; Genentech, South San Francisco, CA, USA) in the study eye at baseline. This was combined with the first sitting of photodynamic therapy (PDT) with verteporfin. The study eye received an anti-VEGF injection at months two and three. All the patients were followed up monthly and the treatment strategy of each patient was reviewed by two senior retina consultants (RS, NFNG).

Distance visual acuity (DVA) was measured using a 4-meter early treatment diabetic retinopathy study (ETDRS) chart. Contrast sensitivity (CS) was measured using the Pelli-Robson chart. CS was recorded based on the contrast of the last group in which two or three letters were correctly read. NVA was recorded using a UiTM Malay related-word reading chart. The UiTM Malay-related word reading chart is a continuous reading chart developed for the Malay-speaking population [13]. The visual functions were tested at baseline before treatment and after six months of treatment.

### 2.2. Image Analysis

Spectral Domain OCT (Spectralis HRA + OCT Heidelberg Engineering Inc., Heidelberg, Germany) was used to evaluate the retinal morphology of the study eyes. OCT was performed with a dilated pupil to achieve the best images. Twenty-five B-scan images with a pattern size of 30 × 20 degrees and distance between B-scan of 242 microns were obtained. All OCT parameter examinations were conducted at baseline prior to commencement of treatment. 

Qualitative and quantitative analyses of OCT images at 1 mm center of the fovea were performed. For qualitative analysis of the OCT image, at first, the fovea was detected and the software caliper (green vertical marker line) was placed at the center of the fovea. Thereafter, a 500 microns section in either direction from the foveal center, that is 1000 microns around the fovea was evaluated. This 1000-micron area was selected manually (Figure 1).

The qualitative parameters that were assessed in 1000 microns (1 mm) within fovea were as follows:

1.Integrity of photoreceptors inner segment and outer segment (IS-OS) junction, external limiting membrane (ELM), and retinal pigment epithelium-Bruch’s membrane (RPE-BM) complex were graded as intact or disrupted. When a 75 percent or more part of ELM, IS-OS junction, and RPE-BM complex 1 mm center of the fovea was present, it was graded as intact. When less than 75 percent of ELM, IS-OS junction, and RPE-BM complex was present, it was graded as disrupted.2.Presence of the following pathologies was recorded 1000 microns (1 mm) within the fovea:Retinal pigment epithelium detachment (PED)—It is the separation of retinal pigment epithelium (RPE) from the RPE-BM complex.Sub-retinal fluid (SRF)—It is seen from the outer border of the RPE to the outer segment of the photoreceptor.Polyp—It is the hypo reflective lumen attached to the posterior surface of PED.

Furthermore, quantitative parameters including thickness and volume of central retinal subfield, central thickness, center maximum, and center minimum thickness were measured in the OCT thickness map) using the incorporated software of Spectralis OCT. When measuring the thickness, the distance between the internal limiting membrane (ILM) and Bruch’s membrane (BM) was considered. In study eyes, the upper marker was automatically placed at the outer border of the ILM and a lower marker was placed at the BM. However, in some of the examined eyes with PED, the lower marker was automatically placed at the RPE. In such cases, the lower marker was adjusted and placed manually at the BM line. In cases where the BM line was missing, the marker was placed considering an approximation of the BM line.

### 2.3. Statistical Analysis

All the data were analyzed using SPSS software for windows, version 17 (IBM SPSS statistics for windows, version 17, New York, NY, USA, IBM corp.). The Shapiro-Wilk test was run to check the normality of the data. The independent t-test and one-way ANOVA between groups were employed to compare the six-month visual functions between the groups divided based on the presence or absence of the baseline qualitative OCT parameters. One Pearson correlation was employed to assess the association between different quantitative OCT parameters and visual functions after six months of treatment.

The study protocol was approved by the Research Ethics Committee of University Kebangsaan Malaysia (UKM/PPI/111/8/NN-186-2014) and the Medical Research and Ethics Committee (MREC), Ministry of Health Malaysia (NMRR-16-1965-31826), which follows the tenants of the Declaration of Helsinki. Written informed consent was obtained from all participants.

## 3. Results

Twenty-six eyes of 26 PCV patients aged between 51 to 88 years were followed up for a period of six months. There were 15 male patients (57.7%) and 11 female patients (42.3%). The mean age of the patients was 68.44 ± 5.48 years. Eyes with SRF within 1000 microns of the fovea at baseline showed significantly worse DVA and CS after six months compared to the eyes without foveal SRF at baseline. Furthermore, eyes with a polyp within 1000 microns of the fovea at baseline showed significantly worse DVA, NVA, and CS after six months compared with the eyes without a foveal polyp at baseline.

However, baseline PED within 1000 microns of the foveal center was not associated with any of the visual functions. Furthermore, mean DVA and CS were significantly better in eyes with a baseline intact ELM within 1000 microns of the foveal center. Similarly, an intact IS-OS junction within 1000 microns of the fovea at baseline exhibited significantly better visual functions after six months in the eyes studied. Moreover, there were significant differences found in the NVA and CS between eyes where both the ELM and IS-OS junction were intact, disrupted or either of them was intact at baseline. Nevertheless, the integrity of the RPE-BM complex showed no association with the visual outcome. (Table 1). There was no correlation between quantitative OCT parameters and six-month visual functions. However, all the final visual functions showed a significant correlation with the baseline visual functions (Refer to Table 2).

## 4. Discussion

The present pilot study reported a significantly worse mean DVA and CS six months after treatment in eyes with foveal SRF, foveal polyp, disrupted ELM, and IS-OS junction at baseline. The presence of foveal polyps and IS-OS junction disruption at baseline significantly decreased the DVA, NVA, and CS six months after treatment.

The presence of foveal SRF resulted in poorer mean DVA and CS after six months of treatment compared to those without foveal SRF at baseline. Similar outcomes in PCV eyes undergoing combination therapy have been reported [14]. Another study has reported that eyes with SRF show a positive effect on treatment outcome but this was in an eye with typical n-AMD [15]. SRF is accumulated between the RPE and neuroretina. As the choroidal neovascular membrane grows, it is often followed by obvious leakage from its small blood vessels [16]. SRF was proved to be consistently associated with a good visual outcome in eyes with typical n-AMD [15] and needed less intensive therapy and extended follow-up intervals [17]. The reason assumed behind this finding was the preservation of the neurosensory retina in these eyes with SRF, compared to those with other associated morphological changes such as IRF or SRT, where the neurosensory retina involved exhibits a more advanced stage of the disease in typical n-AMD [18]. However, different pathophysiology in PCV may result in the difference in visual outcome with foveal SRF. Distinct histopathological features of PCV include dilated vascular channels with venules knotted with choroidal arterioles, serosanguinous RPE detachment associated with extensive sub-RPE choroidal neovascularization leaking into sub-RPE and/or the subretinal space. In later stages, subretinal hemorrhage and fibrosis may be observed [19,20,21,22]. This indicates PCV is a disease that originates below the RPE basal membrane, is confined to RPE in initial stages, and eyes without SRF might have PED and other features confined to the BM and RPE. Whereas PCV eyes with SRF indicate the disease progression above RPE towards the neurosensory retina. Possibly, this could be the reason behind eyes with foveal SRF showing poorer visual outcome compared to eyes without SRF in PCV eyes. However, the presence of foveal PED did not show any association with visual outcome in the present study. This is similar to the previous study reports that did not find any association of PED with high contrast DVA in both PCV eyes undergoing combination therapy [14] and in typical n-AMD [23].

Furthermore, in the present study, eyes with a polyp at the fovea showed comparatively less improvement in the DVA, NVA, and CS compared to the eyes without a foveal polyp. This corresponds to the previous research that reported when eyes have abnormal choroidal vasculature at the foveal avascular zone, it was a negative predictor for visual outcome in PCV eyes undergoing combination therapy [14]. Similarly, a sub-foveal or jaxtra foveal polyp can cause more damage to photoreceptors in eyes undergoing PDT treatment [9]. The presence of a polyp at the fovea indicates an active disease lesion at the foveal area. As the fovea is the most important area responsible for vision, it is justified to consider its role in the visual outcome.

In addition, in the present study, the integrity of the ELM and IS-OS junction has shown significant association with the visual outcome of PCV eyes that have undergone combination therapy of anti-VEGF and PDT. Six months after treatment, the visual parameters including DVA and CS were significantly better in eyes where the ELM was intact at baseline compared to those with disrupted ELM. The ELM is considered a vital structure for the endurance of photoreceptor cells, and the integrity of the ELM is essential for normal visual function. A linear and confluent ELM, located at the border between cell bodies and the inner segment of photoreceptors, is composed of junctional complexes between Muller cells and photoreceptors [24]. It acts as a link between Muller cells and photoreceptors. Muller cells are thought to play an important role in maintaining retinal function. They adjust neural activity by controlling the extracellular concentration of neuroactive substances [25]. Thus, when the ELM is disrupted; the link between Muller cells and the photoreceptor is disturbed which results in structural and functional dysfunction of photoreceptors leading to impairment of vision [26,27,28,29,30]. Recently, it has been reported that the integrity of the ELM can represent the visual status of PCV eyes [3]. Furthermore, previous research has reported the ELM status to be an important predictor of treatment outcome in retinal conditions. The majority of eyes with an intact ELM and a disrupted IS-OS junction finally achieved a better visual and morphological outcome in retinal repair surgery [27,28]. Another study has reported a similar finding where the integrity of the ELM emerged as the only predictor of visual outcomes in patients undergoing anti-VEGF treatment in wet AMD [31]. In correspondence, the integrity of the ELM has been shown to be the only morphological predictor of the best corrected visual acuity (BCVA) in baseline, six-month, and 12-month follow-ups of 20 typical n-AMD patients undergoing anti-VEGF injections [32]. Likewise, in the present study, an intact ELM at baseline showed a better treatment outcome of all visual functions in PCV eyes.

Similarly, the IS-OS junction also termed as ellipsoid zone (EZ), which is mainly formed with the mitochondria within the ellipsoid zone of the outer part of the photoreceptors’ inner segments [3,24,29,33] plays a major role in visual function representing the photoreceptors’ function. The integrity of the IS-OS junctions at baseline was reported as a positive predictive factor for visual outcome in n-AMD eyes undergoing three monthly injections of intravitreal anti-VEGF [34]. Similarly, there is a significant correlation between IS-OS junction integrity and BCVA in n-AMD patients at baseline, and after six months and 12 months of treatment [32]. Likewise, the IS-OS junction along with the ELM and pre-treatment BCVA has been reported to predict 37% of visual outcomes in n-AMD treatment patients [31]. Corresponding to the previous results of n-AMD, the present study reported an intact IS-OS junction at baseline to be associated with better visual prognosis in PCV eyes treated with combination therapy.

Furthermore, eyes with both the ELM and IS-OS junction intact within 1000 microns of the fovea at baseline showed significantly better NVA and CS compared to eyes where either or both were disrupted or missing. It has been suggested that discontinuation of both the ellipsoid zone and the ELM at the fovea indicates that the morphological alterations of the photoreceptor layer may be extended up to the photoreceptor cell bodies and Müller cell cone including changes in the inner and outer segment at the foveola [27,28]. Thus, eyes with both ELM and IS-OS junction disruption indicate more damage than IS-OS junction damage alone. Therefore, eyes with an intact ELM may extend a better visual outcome after treatment.

Although none of the quantitative OCT parameters showed a statistically significant correlation with the six months after-treatment visual outcome, all baseline visual functions including baseline DVA, CS, NVA, showed significant correlation with six months DVA, CS, and NVA. This corresponds to the previous research that showed final DVA to be significantly correlated with the baseline DVA in PCV eyes [9,10,14].

The major strength of the present pilot study is that it has used comprehensive visual functions including DVA, CS, and NVA simultaneously to specifically assess the OCT-based predictor that may be missed by high contrast DVA alone. Another major strength of the study was to use OCT parameters as predictive factors that can be easily adopted by the clinicians in their practice. Previous studies on PCV mainly employed ICGA-based parameters as predicting factors. Considering there is less availability of ICGA, an OCT-based diagnosis of PCV is already recognized. OCT-based parameters that can be associated with the visual outcome will further help in the management of PCV eyes in clinical settings where ICGA may not be available. However, the major limitation of the study was the limited sample size especially while comparing the groups.

## 5. Conclusions

The present pilot study was the first to determine OCT-based factors that might influence the treatment outcome of combination therapy in PCV patients. In spite of the limited study sample, several qualitative OCT parameters such as the presence of a foveal polyp, foveal SRF, disrupted ELM, photoreceptors IS-OS junction showed a negative impact on the six month visual functions in the study eyes. This indicates the fact that those baseline parameters may influence the treatment outcome of PCV eyes. However, further research with increased sample size and study duration is recommended to enhance our understanding of OCT parameters that may assist clinicians in treatment and monitoring PCV eyes treated with combination therapy.

## Figures and Tables

**Figure 1 ijerph-18-05378-f001:**
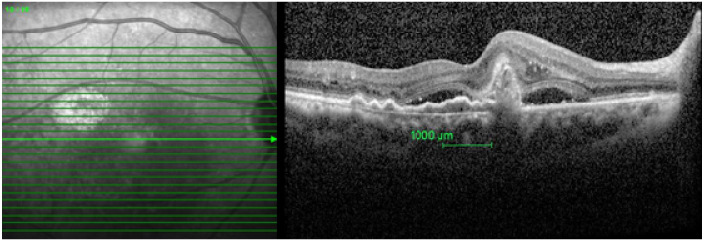
Central 1000-micron area selected in a study eye.

**Table 1 ijerph-18-05378-t001:** Association of qualitative optical coherence tomography (OCT) parameters with visual functions six months after treatment.

Variable	Mean Distance Visual Acuity (logMAR)	*p*	Mean Near Visual Acuity (logMAR)	*p*	Mean Contrast Sensitivity (log Contrast Sensitivity)	*p*
**Foveal Sub -Retinal Fluid**						
Present (*n* = 20)	0.50 ± 0.07	0.043	0.46 ± 0.26	0.202	0.98 ± 0.29	0.040
Absent (*n* = 6)	0.25 ± 0.27		0.30 ± 0.28		1.19 ± 0.21	
**Foveal Polyp**						
Present (*n* = 8)	0.64 ± 0.39	0.011	0.58 ± 0.30	0.032	1.26 ± 0.19	0.005
Absent (*n* = 18)	0.36 ± 0.14		0.35 ± 0.22		0.99 ± 0.24	
**Foveal Pigment Epithelium Detachment**						
Present (*n* = 19)	0.46 ± 0.30	0.662	0.40 ± 0.28	0.477	1.12 ± 0.20	0.898
Absent (*n* = 7)	0.41 ± 0.15		0.48 ± 0.22		1.13 ± 0.32	
**External Limiting Membrane**						
Intact (*n* = 13)	0.34 ± 0.14		0.32 ± 0.14	0.053	1.25 ± 0.20	0.005
Disrupt (*n* = 13)	0.54 ± 0.34	0.044	0.52 ± 0.34		0.99 ± 0.24	
**Photoreceptors Inner segment and outer segment Junction**						
Intact (*n* = 13)	0.34 ± 0.13	0.040	0.31 ± 0.19	0.024	1.23 ± 0.20	0.023
Disrupt (*n* = 13)	0.55 ± 0.32		0.53 ± 0.28		1.01 ± 0.26	
**Retinal Pigment Epithelium-Bruch membrane complex**						
Intact (*n* = 8)	0.48 ± 0.12	0.662	0.55 ± 0.27	0.106	1.12 ± 0.34	0.662
Disrupt (*n* = 18)	0.42 ± 0.15		0.37 ± 0.24		1.24 ± 0.25	
**External Limiting Membrane + Photoreceptors Inner Segment and Outer Segment Junction**						
Both intact	0.32 ± 0.13	0.086	0.27 ± 0.19	0.040	1.27 ± 0.19	0.029
Either intact	0.47 ± 0.90		0.55 ± 0.17		1.12 ± 0.15	
Both disrupted	0.57 ± 0.35		0.52 ± 0.29		0.98 ± 0.26	

**Table 2 ijerph-18-05378-t002:** Correlation between quantitative OCT paramaters and visual functions six months after treatment.

Variable	Average Retinal Thickness	Average Retinal Volume	Central Thickness	Maximum Thickness of Central 1 mm	Minimum Thickness of Central 1 mm	Baseline Distance Visual Acuity/Baseline Near Visual Acuity/Baseline Contrast Sensitivity/Baseline Reading Speed
**6 months distance visual acuity**	*r* = 0.219 CI (0 .63 to −0.19) *p* = 0.282	*r* = 0.213 CI (0.64 to −0.19) *p* = 0.295	*r* = 0.339 CI (0.74 to −0.05) p = 0.90	*r* = 0.215 CI (0.62 to −0.19) *p* = 0.292	*r* = 0.040 CI (0.46 to −0.38) *p* = 0.848	*r* = 0.53 CI (0.88 to 0.17) *p* = 0.006
**6 months near visual acuity**	*r* = 0.031 CI (0.45 to −0.39) *p* = 0.880	*r* = −0.247 CI (0.16 to −0.65) *p* = 0.223	*r* = 0.145 CI (0.56 to −0.27) *p* = 0.480	*r* = 0.086 CI (0.56 to −0.34) *p* = 0.676	*r* = −0.133 CI (0.28 to −0.55) *p* = 0.517	*r* = 0.40 CI (0.78 to 0.01) *p* = 0.042
**6 months contrast sensitivity**	*r* = −0.131 CI (0.28 to −0.55) *p* = 0.523	*r* = 0.302 CI (0.70 to −0.10) *p* = 0.134	*r* = −0.206 CI (0.20 to −0.61) *p* = 0.312	*r* = 0.004 CI (0.42 to −0.41) *p* = 0.984	*r* = −0.002 CI (0.42 to −0.42) *p* = 0.994	*r* = 0.41 CI (0.79 to 0.02) *p* = 0.037

*r* = correlation coefficient, *p* = statistical significance.

## Data Availability

Not applicable.

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
