# Peer review of "Baseline Optical Coherence Tomography Parameters That May Influence 6 Months Treatment Outcome of Polypoidal Choroidal Vasculopathy Eyes with Combination Therapy: A Short-Term Pilot Study"

_ijerph, 2021, doi:10.3390/ijerph18105378_

Round 1

Reviewer 1 Report

Introduction: For the non-expert reader it would be helpful to introduce working principles and details for clinical use within the introduction. Same applies for PCV.

Methods: Include details of inclusion and exclusion criteria. For example, you mention that larger lesions and thinned foveas were excluded. What were the exact exclusion dimensions and when and how was it assessed?

Please provide details on how you determined the sample size.

It is not clear how you determined the qualitative parameters. It is not clear that you use those parameters for grouping. You would need to consider those groups in your sample size analysis, otherwise (as it is in your study) you have unequal sample sizes for each of the groups.

Fig. 2 is squeezed and not neccessary you can refer to the handbook.

Your statistical analysis is worded wrongly.  If you compare baseline measures with 6 months follow up measures, it is a dependent sample comparison. However, you did compare groups, which were independent. Please clarify this section. Which type of ANOVA did you use? You would need to include normality testing if you want to use it. 

Results: The significant comparisons between the two groups are only significant in those with less than 10 comparisons. Please provide units for your outcome measures.

Discussion: Your disussion is good. However, the conclusion you draw from your results are too powerful. In some of the groups you only have 6 patients.

This is a very interesting and important study and you should submit the results of the actual study (for which this submission is a pilot project).

Author Response

Dear Reviewer,

Thank you for your comments. I have addressed all the comments as stated below.

  1. Introduction:

Comment: For the non-expert readers it would be helpful to introduce working principal and details of clinical use within introduction. Same applied to PCV

Response: Introduction to clinical use of OCT has been incorporated in the introduction. Brief introduction about PCV has also been incorporated.

“Ophthalmic imaging has been revolutionized by introduction of Optical Coherence Tomography (OCT). It uses interferometric imaging technique that produces cross-sectional images by mapping reflections of low-coherence laser light from tissue based on its depth. From the time of its inception, OCT has been efficiently attempting to provide minute clinical measurements of retinal layers in an automated fashion. With OCT, imaging of multilayer retina represents depth of each layer with amplitude of spectrum modulation that is proportional to the specific reflectivity of the layer. Thus, OCT plays major role in the diagnosis and management of a broad range of retinal diseases enabling a detailed evaluation of conditions including age related macular degeneration, central serous retinopathy, retinal vein occlusion, diabetic retinopathy and inherited retinal diseases [1]. With time, newer approaches towards identifying different OCT derived parameters have been tested for retinal pathologies. Eventually, several OCT based qualitative and quantitative parameters have been adopted in diagnosis, treatment and monitoring of different retinal diseases including different sub types of neovascular age related macular degeneration (AMD) [2-4].”

“Polypoidal Choroidal Vasculopathy (PCV) is a commonly seen subtype of neovascular age related macular degeneration (n-AMD) in Asian population including Malaysian population [5,6]. It is characterized by multiple pigment epithelium detachment. polypoidal lesions, brunch vascular network (3). Intravitreal anti-VEGF and PDT are the mainstay of treatment on PCV eyes. Visual functions, retinal morphology and quality of life significantly improve with treatment in PCV eyes [7,8].

  1. Methods:

Comments: Include details of inclusion and exclusion criteria. For example, you mention that larger lesion and thinned fovea. What were the exact dimensions and when and how it was measured?

Response: Larger lesion was considered as lesion size > 5400 μm with ICGA and thinned fovea was considered as central retinal thicknesss < 200 μm with OCT. All the inclusion and exclusion criteria were evaluated by senior retina consultants before the recruitment of the subjects. The diagnosis of PCV has already been discussed in our earlier study. The following has been incorporated in the manuscript.

“PCV patients undergoing other treatment for any retinal pathology other than PCV and any history of previous treatment (anti-VEGF injection, PDT, or laser) for PCV were excluded. Similarly, patients with larger lesion (lesion size > 5400 μm with ICGA), multiple polyps, foveal fibrosis and thinned fovea (central retinal thicknesss < 200 μm with OCT) wherein PDT is not advisable, were excluded (12). Furthermore, eyes with significant media opacity were also excluded. All patients underwent comprehensive eye examination and Ophthalmic imaging including FFA, OCT and ICGA. Senior retinal consultants (N.F.N, R.A.A.A.) evaluated all the subjects in details before recruitment. The diagnosis of PCV has been described in our earlier paper [3].”

Comment: Provide details how you determined sample size.

Response: In this pilot study, all patients with PCV who were seen in the ophthalmology department of this public hospital from December 2016 to March 2017 and who met the inclusion and exclusion criteria of the study were recruited.

However, the unequal and lower sample size has been put under the limitation of the study.

Comments: It is not clear how you determined the qualitative parameters. It is not clear that you used those parameters for grouping. You need to consider those groups in sample size analysis. Otherwise, you will have unequal sample size in each group.

Response: The qualitative parameters were determined using OCT. The detailed description of the same has been incorporated in the manuscript. The unequal and lower sample size has been put under the limitation of the study.

“Qualitative and quantitative analysis of OCT images at 1 mm centre of the fovea were performed. For qualitative analysis of OCT image, at first, fovea was detected and the software caliper (green vertical marker line) was placed at the centre of the fovea. Thereafter, 500 microns section in either direction from the foveal centre that is 1000 microns around fovea was evaluated. This 1000-micron area was selected manually (Figure 1).

The qualitative parameters that were assessed in 1000 micron (1mm) of within fovea were as follows:

  1. Integrity of photoreceptors inner segment and outer segment (IS-OS) junction, external limiting membrane (ELM), and retinal pigment epithelium-Bruch’s membrane (RPE-BM) complex were graded as intact or disrupted. When 75 percent or more part of ELM, IS-OS junction, RPE-BM complex of 1 mm centre of fovea were present, it was graded as intact. When less than 75 percent of ELM, IS-OS junction, RPE-BM complex was present, it was graded as disrupted.
  2. Presence of the following pathologies were recorded in 1000 micron (1mm) of within fovea:
    1. Retinal pigment epithelium detachment (PED) - It is the separation of retinal pigment epithelium (RPE) from RPE-BM complex.
    2. Sub-retinal fluid (SRF) - It is seen from outer border of RPE to outer segment of photoreceptor.
    3. Polyp – It is the hypo reflective lumen attached to the posterior surface of PED.”

Comment: Figure 2 is squeezed and not necessary. It can refer to the handbook.

Response:  Reviewer 2 has suggested to retain Figure 2 but incorporate CI. Therefore, Figure 2 has been retained. But if both reviewers feel that Table 2 should be removed, I will remove it.

Comments: Your statistical analysis is worded wrongly. If you compare baseline measures with 6 months follow up, it is a dependent sample comparison. However, you did compare groups which were independent. Please clarify this section. Which type of ANOVA have you used? You need to include normality testing if you want to use that.

Response: Statistical analysis has been rewritten. Normality test employed is incorporated.

“All the data were analysed using SPSS software IMB SPSS 17; SPSS Inc. USA. Shapiro-Wilk test was run to check the normality of the data. Independent-t test and one way between groups ANOVA were employed to compare the 6 months visual functions between the groups divided based on the presence or absence of the baseline qualitative OCT parameters. Pearson’s correlation was employed to assess the association between different quantitative OCT parameters and visual functions after 6 months treatment”

  1. Results:

Comment: The significant comparison between the 2 groups are only significant in those less than 10 comparisons. Please provide units of your outcome measures

Response: Units have been incorporated. Sample size has been mentioned as limitation. However, in spite of limited sample, some of the parameters showed significant impact on the treatment outcome.

  1. Discussion:

Comment: Your discussion is good. However, the conclusions you draw from your results are too strong. In some of the groups you have 6 patients.

Response: Conclusion has been rewritten.

“The present pilot study was the first to determine OCT based factors that might influence the treatment outcome of combination therapy in PCV patients. In spite of limited study sample, several qualitative OCT parameters such as presence of foveal polyp, foveal SRF, disrupted ELM, photoreceptors IS-OS junction showed a negative impact on 6 months’ visual functions in the study eyes. This indicates towards the fact that those baseline parameters may influence the treatment outcome of PCV eyes. However, further research with increased sample size and study duration is recommended to enhance our understanding on OCT parameters that may assist clinicians in treatment and monitoring PCV eyes treated with combination therapy.”

Reviewer 2 Report

I consider this study to have valuable data that would be of interest if published. However it needs a major revision. In details: 1. The materials and methods section should be re-written. Please divide this section only into three paragraphs i.e. study  design and treatment (including sample size and visual functions), image analysis (including OCT parameters) and statistical analysis. Than put the consent information and ethic approval at the end of this section. 2. Please write the results sections as one paragraph and re-write the Table 2 - the retinal parameters should be given with their 95% CI 3. The discussion section is too long. However I like the conclusions, there are too many hypotheses that are not supported by the study results, so please shorten it. 

Author Response

Dear Reviewer,

Thank you for your comments. I have addressed all your comments as below.

General comment: I consider this study to have valuable data that would be of interest if published. However, it needs a major revision.

  1. Comment: Materials and method section should be re written. Please divide the sections into three paragraphs. i. e study design and treatment (including sample size and visual functions), image analysis (including OCT parameters), statistical analysis. Than put the consent information and ethical approval at the end of this section.

Response: Materials and methods section has been re-written as per the adviced. The consent information and ethical approval has been moved to the end of this section.

  1. Comment: Please write the result sections as one paragraph and re-write the table 2 - the retinal parameters should be given with 95%CI.

Response: Result section has been put in one paragraph as advised. Table 2 has been re-written with CI for each of the parameters.

  1. Comment: Discussion section is too long. However, I like the conclusion, there are many hypothesis that are not supported by the study results. So please shorten it.

Response: Discussion has been shortened. Considering the limited sample size, part of the discussion has been removed.

Round 2

Reviewer 1 Report

Thank you for making the recommended changes. I recommend to leave table 2 in.

Reviewer 2 Report

I am satisfied with the authors' reply